# A Stochastic Analysis of the Effect of Trading Parameters on the Stability of the Financial Markets Using a Bayesian Approach

Rolando Rubilar-Torrealba [1,*,†] , Karime Chahuán-Jiménez [2,†] and Hanns de la Fuente-Mella [3,†]

1   Universidad de La Frontera, Facultad de Ciencias Jurídicas y Empresariales, Departamento de Administración y Economía, Temuco 4811230, Chile
2   Centro de Investigación en Negocios y Gestión Empresarial, Escuela de Auditoría, Facultad de Ciencias Económicas y Administrativas, Universidad de Valparaíso, Valparaíso 2362735, Chile; karime.chahuan@uv.cl
3   Instituto de Estadística, Facultad de Ciencias, Pontificia Universidad Católica de Valparaíso, Valparaíso 2340031, Chile; hanns.delafuente@pucv.cl
*   Correspondence: rolando.rubilar.t@gmail.com
†   These authors contributed equally to this work.

**Abstract:** The purpose of this study was to identify and measure the impact of the different effects of entropy states over the high-frequency trade of the cryptocurrency market, especially in Bitcoin, using and selecting optimal parameters of the Bayesian approach, specifically through approximate Bayesian computation (ABC). ABC corresponds to a class of computational methods rooted in Bayesian statistics that could be used to estimate the posterior distributions of model parameters. For this research, ABC was applied to estimate the daily prices of the Bitcoin cryptocurrency from May 2013 to December 2021. The findings suggest that the behaviour of the parameters for our tested trading algorithms, in which sudden jumps are observed, can be interpreted as changes in states of the generated time series. Additionally, it is possible to identify and model the effects of the COVID-19 pandemic on the series analysed in the research. Finally, the main contribution of this research is that we have characterised the relationship between entropy and the evolution of parameters defining the optimal selection of trading algorithms in the financial industry.

**Keywords:** cryptocurrencies; econometric models; stochastic processes; Bayesian analysis; market efficiency; entropy

## 1. Introduction

The concept of cryptocurrency started to become popular after Bitcoin in 2008 [1], achieving, through a cryptographic process, the issuing and transferring of digital tokens across a digital communication infrastructure integrated by people connected in a network [2].

The growth experienced by the cryptocurrency market outpaces any other market. Furthermore, this growth has not been affected in the past by events that affected economic transactions [3], such as mining operations, environmental impact, or Elon Musk's notorious refusal to own cryptocurrencies. Moreover, these effects partly explain why they have been considered a haven [4–6]. The Chinese government's mining ban and restrictions imposed in other countries were due to the environmental impact. Cryptocurrency mining has a huge environmental impact in Asia, Latin America, and Africa [7–9].

On the other hand, the development of the COVID-19 pandemic made the cryptocurrency market unpredictable, generating an abnormal increase in demand [5,10–12], causing abnormal volatility which generated different expectations from investors [13]. The volatility of this type of asset generates a need to measure its associated risk.

This research focuses on identifying entropy states by selecting optimal parameters using a Bayesian approach in the high-frequency trade of the cryptocurrency market, especially in Bitcoin. Thus, this paper adds value to the specialised literature regarding the relationship between the Bitcoin return series's entropy and the selection of the optimal parameters estimated, leading the research question towards the potential relationship between the concept of entropy applied to finance.

The sections of this paper are organised as follows: Firstly, Section 1 introduces the area of research, while Section 2 reviews the relevant literature. Secondly, Section 3 presents the materials and methods used in the research. Section 4 provides the results obtained, while Section 5 contains a discussion of those results. Finally, Section 6 presents the conclusions drawn from the research.

## 2. Literature Review

The better performance of cryptocurrencies is shown through higher levels of liquidity in periods of market stress [14]. Investors tend to favour well-known virtual currencies over lesser-known ones during periods of high volatility. This phenomenon can be attributed to investors' firm and persistent herding behaviour, a tendency to follow the positive or negative trends set by Bitcoin, Ethereum, and other highly capitalised cryptocurrencies [15].

Trabelsi's research findings [16] indicated a lack of substantial transmission of shocks between the emerging cryptocurrency market and other financial markets. According to their suggestions, cryptocurrencies are autonomous financial assets that should not pose a threat to the stability of the financial system. Regarding the connection within cryptocurrency markets, the time-frequency-dynamic link is based on a natural condition. Furthermore, the spread decomposition over the index is predominantly influenced by a component with a short frequency (2–4 days), concluding that this emerging market is characterised by a high degree of speculation. The results support the decision-making of regulators and investors.

According to [17], the Bitcoin price forecasting model is one of the most famous mathematical models in financial technology due to its hefty price fluctuations and complexity. Bitcoin was the pioneering cryptocurrency developed using blockchain, cryptography, and peer-to-peer technology. Numerous mathematical models have been developed within financial technology to predict the future price of Bitcoin. These mathematical models assist investors in making informed decisions and optimising their investments.

Before delving into the process by which the price of Bitcoin is determined, it is important to establish whether it is a haven or a risky asset. Proponents argue that Bitcoin's decentralised nature and limited supply of 21 million coins, much like gold, make it an effective hedge against inflation and a safe-haven asset. Conversely, some contend that the market for Bitcoin is characterised by high speculation, with prices showing a positive correlation with various risk assets, indicating that it is a risky market [18].

The research results of [19] demonstrate that their system can accurately forecast cryptocurrency price trends, generate profitable trades, and, in most cases, outperform the basic buy-and-hold strategy. Trading XRP achieved the best performance compared to Binance Coin, Ethereum, and Bitcoin [19]. For all coins, the system predicted better long-term trends than short-term trends. Ref. [20] attempted to predict cryptocurrency price fluctuations by analysing online community comments. They found that positive comments significantly affected Bitcoin price fluctuations, whereas negative remarks significantly impacted the prices of other cryptocurrencies, such as Ripple (XRP) and Ethereum. The relationship between returns, subjectivity, and Twitter polarisation was realised by [21] and reported significant cross-correlation values between online search volume and Bitcoin trading volume [22].

Whether Bitcoin's predictable price behaviour interests market participants, predictability is incompatible with efficient market hypotheses for any financial instrument [23]. In recent years, ref. [24] used many tests to analyse the efficiency of the Bitcoin market and concluded that Bitcoin becomes much more efficient in the most recent time series sample;

ref. [25] used eight tests on an odd integer power transformation of Bitcoin returns, which revealed that the returns exhibit a lack of efficiency, ref. [26] examined the long-range dependence properties of Bitcoin's price dynamics, reporting a trend in the direction of greater efficiency; similar to the results of [27].

On the other hand, ref. [28] assessed the efficiency of Bitcoin relative to other assets, such as gold, stocks, and currency. Their results showed that the Bitcoin market could be more efficient, supporting the findings of [29].

The findings offer several implications for understanding Bitcoin hedging and diversification properties. The research of [30] concluded that there is a two-way relationship between Bitcoin and green bonds [31]. The value of green bonds tends to increase in response to a positive shock in the Bitcoin market, and conversely, a negative shock in Bitcoin decreases the value of green bonds. Additionally, the research found a direct correlation between clean energy, oil prices, and green bonds. This suggests that green bonds do not constitute a separate asset class but rather reflect the performance of other asset classes such as Bitcoin, clean energy, and oil prices. Furthermore, ref. [32] found that while cryptocurrencies are strongly related to returns and volatility, they are less related to liquidity, indicating a gradual increase in the importance of privacy-oriented cryptocurrencies, similar to that described by [33].

Additionally, ref. [34] conducted an assessment of the high-frequency asymmetric volatility link between precious metal markets and Bitcoin. Their findings revealed that spillovers are susceptible to negative shocks and political events while reaffirming the critical asymmetrical nature of volatility connectivity. Ref. [35] investigated the high-frequency volatility and return spillovers between cryptocurrencies and discovered that the spillover patterns for returns and volatility vary among cryptocurrencies, and [36,37] investigated the high-frequency link between Bitcoin and other cryptocurrencies that experienced high transaction volumes during the COVID-19 pandemic, determining that a positive transmission effect exists within the cryptocurrency markets. From [38], the efficiency of the Bitcoin market was reported to be lower than that of other financial markets. However, there is no notable difference between Bitcoin and other markets regarding their long-term market equilibrium.

According to [39], the incorporation of blockchain technologies, which facilitate the integration of cryptocurrencies into daily life and the economy, carries inherent risks. As a result, regulators are constantly focused on maintaining stability, which often involves a cautious approach. To achieve this objective, it is essential to establish regulations and guidelines that can be applied at both national and international levels. Additionally, according to [40], the market participants differentiate similar financial assets using blockchain technology.

According to [41], there exists a correlation between the entropy of intra-day returns and the daily exchange rate of Bitcoin. A positive correlation is observed between the daily log price of Bitcoin and the entropy of intra-day returns; this suggests that entropy serves as a predictive indicator for Bitcoin price dynamics. Furthermore, the positive and statistically significant entropy coefficient implies that market uncertainty may be a driving factor in shaping the dynamics of Bitcoin prices.

Conducting an entropy analysis of both the training and test samples unveiled characteristics such as extensive memory, high stochasticity, and topological complexity [42]. The development of a non-linear dynamic present in the Bitcoin time series is rationalised by machine learning techniques. Optimal parameter values for support vector regressions (SVR), Gaussian and Poisson regressions (GRP), and k-nearest neighbours (kNN) are determined through the use of Bayesian optimisation. Through the evaluation of multiple performance metrics, the findings indicate that Bayesian regularisation (BRNN) yields outstanding forecasting precision while maintaining unhindered and notably rapid convergence.

## 3. Materials and Methods

### 3.1. Capital Markets Modelling

Stock markets are affected by a large number of factors, events, and variables that imply a high complexity and non-linearity in the relationship of these components. Therefore, it is challenging to forecast future stock trends for investors. Methodologically, the literature has evaluated multiple forecasting methods to predict stock prices from the origin of the stock markets [43].

Predictions of an asset's price volatility can be made by utilising historical price data and/or information derived from options prices. Thus, in recent years, some important techniques have been developed for asset transactions in the capital markets, which seek to maximise the efficiency of trading strategies by seeking out the opportunities offered by the market. Financial researchers have proposed numerous forecasting models, including Box and Jenkins' autoregressive moving average (ARMA) model [44], the autoregressive integrated moving average model (ARIMA), Engle's autoregressive conditional heteroscedasticity (ARCH) model [45], and Bollerslev's generalised ARCH (GARCH) model [46].

Conventional time series models frequently implement AR, a widespread and significant method for time series forecasting [47]. Nonetheless, the AR technique's capacity for modelling time-series data is limited. For this research, we will use an approach that investment managers widely use in their active operations, the moving average model ($MA(q)$), which has presented good results in the forecasting of series in the capital market.

The moving average model attempts to capture the stable behaviour of financial series by averaging over the last $q$ observations. This approach allows us to have a simple tool to forecast the stable behaviour of the time series, characterised as

$$y_t = c_0 + \varepsilon_t + \theta_1 \varepsilon_{t-1} + \theta_2 \varepsilon_{t-2} \ldots + \theta_q \varepsilon_{t-q}, \tag{1}$$

where $y_t$ corresponds to the price of the financial asset in period $t$; $c_0$ corresponds to a constant of the process; $\theta_s$ corresponds to the parameter accompanying the error term of period $t-s$; $\varepsilon$ corresponds to the error term in period $t-s$; and $\varepsilon_{t-s}$ corresponds to the error term in period $t-s$.

For the simplest and most widespread case in financial operations, we can define the same weight of the lags of the errors with a value of $\theta = 1/q$ and assume the value of the process constant equal to zero, which transforms Equation (1) into

$$y_t = \varepsilon_t + \frac{1}{q} \sum_{i=1}^{q} \varepsilon_{t-i}. \tag{2}$$

If we assume the expected value of $\varepsilon_t = 0$, then the forecast of future price levels simply corresponds to the average of the last $q$ periods. Figure 1 shows part of the price series with the two moving average series of values $q_1 = 32$ and $q_2 = 46$ days, respectively.

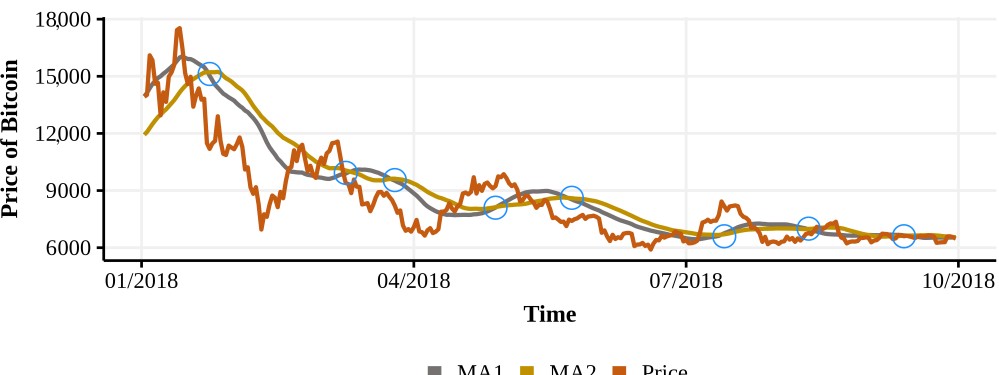

**Figure 1.** Bitcoin price series and two moving average samples.

The points marked with circles correspond to moments in time when the moving averages cross. These crossings of the moving averages mark the buy and sell signals in the stock market, points at which profits can be maximised if the selection of the moving averages is appropriate.

Two different trading mechanisms are proposed for this research, depending on the generated signal. The first mechanism corresponds to the generation of a buy signal when the shorter moving average surpasses, the longer moving average, and a sell signal is produced when the opposite occurs (TR1). The second mechanism corresponds by generating a buy signal when the shorter moving average falls below the longer moving average, and a sell signal is produced conversely (TR2).

### 3.2. Approximate Bayesian Computation Estimation

In this section, we will proceed to estimate the parameters that define the behaviour of the moving average trading parameters ($MA(q_1)$, $MA(q_2)$), utilizing techniques based on Bayesian inference.

In Bayesian analysis, the unknown parameter is represented as a random variable $\theta$ with a probability distribution $\pi(\theta)$, known as the prior distribution. The prior distribution encapsulates our understanding of the moving average trading parameters' value.

Given a parameter $\theta$, the observed data $X$ is assumed to follow a density function denoted as $f(x/\theta)$, where $f(x/\theta)$ defines a parametric model with $\theta$ as its parameter. The joint distribution of $\theta$ and $X$ is then the product $f(x, \theta) = f(x/\theta)\pi(\theta)$, and the marginal density of $X$ (in the continuous case) is $f(x) = \int f(x, \theta)d\theta = \int f(x/\theta)f(\theta)d\theta$.

The conditional density of $\theta$ given the observed data are $f(\theta/x)$. The distribution of $\theta$ after considering the observed data are referred to as the posterior distribution. It is typically represented as follows

$$\text{Posterior density} \quad \propto \quad \text{Likelihood} \times \text{Prior density} \tag{3}$$

In order to proceed with a Bayesian strategy for the estimation of $MA(q_1)$, $MA(q_2)$, we have as an alternative the approximate Bayesian computation (ABC) technique, which corresponds to a series of acceptance–rejection algorithms that use a set of summary statistics from a random sample to compute the posterior distribution. In ABC techniques, the likelihoods are replaced by a simulation procedure, allowing greater flexibility in dealing with complex problems.

The natural strengths of the ABC technique are its ease of programming, the production of independent observations and the adaptability of the technique to a variety of circumstances, allowing it to address very complex problems that cannot be solved by other techniques.

Several alternatives for obtaining a posterior distribution using ABC techniques are described in [48]. Alternatives described in the existing literature can be found in [49–52], to name a few.

The general Algorithms 1 and 2 were used for the TR1 and TR2 trading algorithms, respectively. ABC algorithms seek to simulate the values of $q_1$ and $q_2$ with a list of possible values that the moving average parameters can take ($list_1$, $list_2$) that allow for profits above an exogenously defined threshold ($minProf$). If the randomly selected parameters allow these desired profits, these values are stored in a vector $Q_1$ and $Q_2$ of size *ndens* each. Finally, the density of this vector is calculated, which corresponds to the posterior distribution of the accepted data.

---

**Algorithm 1** ABC algorithm for modelling MA1 and MA2 parameters (TR1)

---

**Require:** Define an initial value $order = buy$. Define an empty vector $p_{sell}$, $p_{buy}$. Define an empty vector $Q1$, $Q2$.

    **while** $i < ndens$ **do**
        **while** $q_1 \leq q_2$ **do**
            Simulate $q_1 \sim rand(list_1)$
            Simulate $q_2 \sim rand(list_2)$
        **end while**
        $MA1 = MA(q_1)$
        $MA2 = MA(q_2)$
        **for** $s = 1 : Time$ **do**
            **if** $MA1[s] > MA2[s]$ and $MA1[s-1] \leq MA2[s-1]$ **then**
                **if** $order == buy$ **then**
                    $order = sell$
                    $p_{sell} = append(p_{sell}, Price[s])$
                **end if**
            **end if**
            **if** $MA1[s]] \leq MA2[s]$ and $MA1[s-1] > MA2[s-1]$ **then**
                **if** $order == sell$ **then**
                    $order = buy$
                    $p_{buy} = append(p_{buy}, Price[s])$
                **end if**
            **end if**
        **end for**
        $profit = \frac{p_{sell} - p_{buy}}{p_{buy}}$
        **if** $profit - minProf > 0$ **then**
            $Q1 = append(Q_1, q_1)$
            $Q2 = append(Q_2, q_2)$
        **end if**
        $i = i + 1$
    **end while**

---

The first part of the algorithms shows a cycle that will end when a minimum number of accepted data is obtained to generate the corresponding densities. For each iteration, the time series will be calculated based on the $q_1$ and $q_2$ parameters obtained randomly, allowing us to calculate the trading signal points. Afterwards, profits will be calculated to compare them with an acceptable minimum level. In case the difference is positive, the simulated parameters will be saved in an array that will end when its size is sufficient to calculate the density of the parameter value.

Figure 2 shows the parameter density of MA1 and MA2 for a given time sample. In this figure, we can observe two clearly defined groupings of data, in which it can be seen that there is no single selection that allows for a certain level of profit but that they are associated with a certain distribution with multimodal characteristics.

The selection of the optimal parameters corresponds to those that maximise the joint probability of obtaining profit above a given threshold, taking into consideration, for more sophisticated applications, the possibility that the selection is completely random conditional on the joint density obtained from the ABC algorithm.

---

**Algorithm 2** ABC algorithm for modelling MA1 and MA2 parameters (TR2)

---

**Require:** Define an initial value *order* = *buy*. Define an empty vector $p_{sell}$, $p_{buy}$. Define an empty vector Q1, Q2.

 **while** $i < ndens$ **do**
  **while** $q_1 \leq q_2$ **do**
   Simulate $q_1 \sim rand(list_1)$
   Simulate $q_2 \sim rand(list_2)$
  **end while**
  $MA1 = MA(q_1)$
  $MA2 = MA(q_2)$
  **for** $s = 1 : Time$ **do**
   **if** $MA1[s] \leq MA2[s]$ and $MA1[s-1] > MA2[s-1]$ **then**
    **if** $order == buy$ **then**
     $order = sell$
     $p_{sell} = append(p_{sell}, Price[s])$
    **end if**
   **end if**
   **if** $MA1[s]] > MA2[s]$ and $MA1[s-1] \leq MA2[s-1]$ **then**
    **if** $order == sell$ **then**
     $order = buy$
     $p_{buy} = append(p_{buy}, Price[s])$
    **end if**
   **end if**
  **end for**
  $profit = \frac{p_{sell} - p_{buy}}{p_{buy}}$
  **if** $profit - minProf > 0$ **then**
   $Q1 = append(Q_1, q_1)$
   $Q2 = append(Q_2, q_2)$
  **end if**
  $i = i + 1$
 **end while**

---

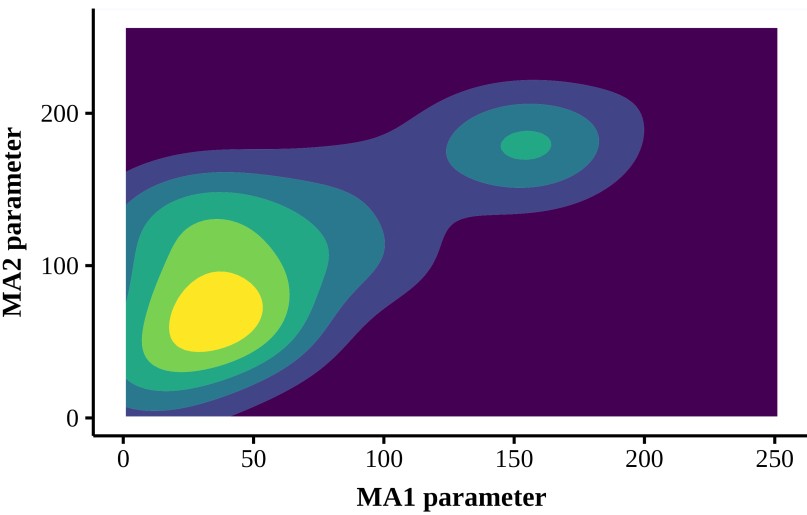

**Figure 2.** Joint density of MA parameters.

### 3.3. Entropy and Finance

Entropy, as a fundamental concept, has the ability to capture and represent various notions such as uncertainty, ambiguity, and disorder that arise in the context of a stochastic process, and it has been widely applied in various scientific fields, such as physical sciences, mathematics and economics, to name a few. Entropy is a widely applicable and versatile concept that is not constrained by the theoretical probability distribution of the random

variables to which it is applied. Various entropy functions have been identified in the field of finance, each based on different axioms imposed on probability distributions. Examples of these functions include Shannon entropy, Renyi entropy, Tsallis entropy, conditional entropy, permutation entropy, approximate entropy, and transfer entropy. These diverse entropy measures have the potential to offer valuable insights for financial analysis [53].

The utilisation of entropy in finance can be viewed as an expansion of information theory and could prove to be a valuable instrument in asset pricing and portfolio selection [53] because entropy represents countable information that can be obtained from the price series observed in the capital markets. The definition of entropy defined for this research corresponds to the one developed by Shannon [54] in their seminal paper, define as:

$$S_n = - \sum_{i=1}^{n} p_i \ln p_i, \tag{4}$$

where $\sum_{i=1}^{n} p_i = 1$, $p_i \geq 0$ and $0 \ln 0 = 0$. However, the identification of entropy can be extended to other definitions, such as Renyi's definition. The entropy definition proposed in this research corresponds to a first approximation helping us understand how the selection of optimal parameters for trading financial assets is linked to the evolution of entropy over time.

*3.4. k-Means Clustering Algorithm*

Another element necessary for this research corresponds to the classification of a finite number of states associated with the densities of the moving average parameters that optimise the trading strategy and the entropy of the Bitcoin price series.

For a dataset $X = \{x_1, x_2, \dots x_N\}$, $x_n \in R^d$, we define k-means clustering algorithm dependents on a set of centroids $m_1, m_2, \dots, m_M$ and a subset $C_k \in C$ which contains $x_i$ as

$$\arg \min_C \sum_{i=1}^{N} \sum_{k=1}^{M} I(x_i \in C_k) ||x_i - m_k||^2, \tag{5}$$

where $I(X) = 1$ if $X$ is true and 0 if not. In particular, we will employ the k-means algorithm to determine the number of states present in the experimental data obtained in this research [55,56].

**4. Results**

*Data*

The data used in this research correspond to the daily prices of the Bitcoin cryptocurrency from May 2013 to December 2021. However, the data used for training is from May 2013 to May 2017, leaving data available to track and test the algorithms from June 2017 to December 2021.

Below, Table 1 shows the parameters defining the behaviour of Algorithms 1 and 2. The $ndens = 100$ value corresponds to the number of values used to calculate the joint density and the elements from experiments that exceed a profit greater than $minProf = 0.05$. The case of the $Time = 500$ parameter implies that we will observe the effect of trading algorithms for the last 500 periods, approximately two years of trading on the capital markets. Finally, the size of $list_1$ and $list_2$ indicate the number of possible moving averages that can be used for the trading algorithms, which in the case of this experiment range from approximately one day to four years of transactions.

**Table 1.** Parameters of the estimation process.

| Variable | Value |
|----------|-------|
| $ndens$ | 100 |
| $Time$ | 500 |
| $minProf$ | 0.05 |
| $list_1$ | $1, 2, \ldots, 999$ |
| $list_2$ | $2, 3, \ldots, 1000$ |

Figure 3 shows the summary of the main results of the optimisation model. The first sub-figure corresponds to the density of the profitability results when applying the best alternative algorithm to perform the trade model. We can observe that the expected profitability is around 9% with minimum values of 5% and maximum values of 14%, considering 500 periods for the evaluation. Note that there are values equal to zero which implies that the optimal decision was to maintain the position during that period of time.

The second sub-figure shows the frequency of use of each type of algorithm, showing that 62% corresponds to the use of Algorithm 1, while 38% corresponds to Algorithm 2, showing a greater predominance of Algorithm 1 as the optimal trade criterion.

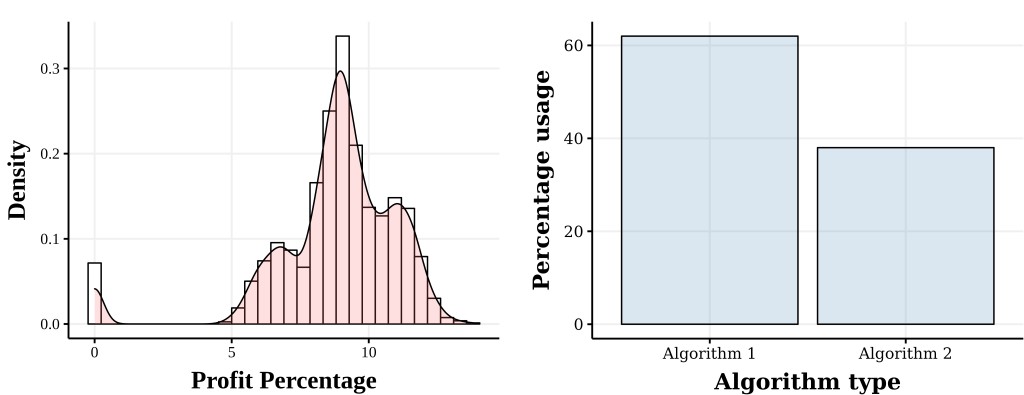

**Figure 3.** Profitability results and percentage use of algorithms.

Figures 4 and 5 correspond to the daily evolution of the optimal MA1 and MA2 mean parameters for the case of the TR1 and TR2 trading algorithms, respectively. The results show the behaviour of the parameters for both trading algorithms, in which sudden jumps are observed that can be interpreted as changes in states of the generated time series. These changes in state can lead to a modification of the use of the algorithm that allows an optimisation of the trading decision.

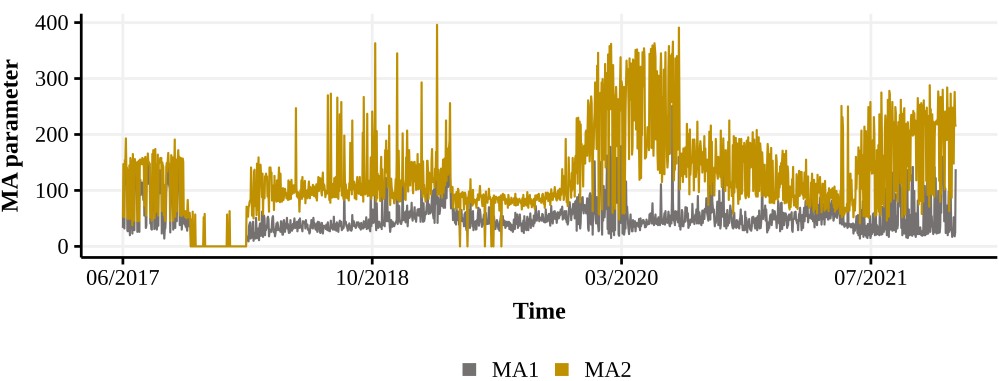

**Figure 4.** Evolution of the MA parameters.

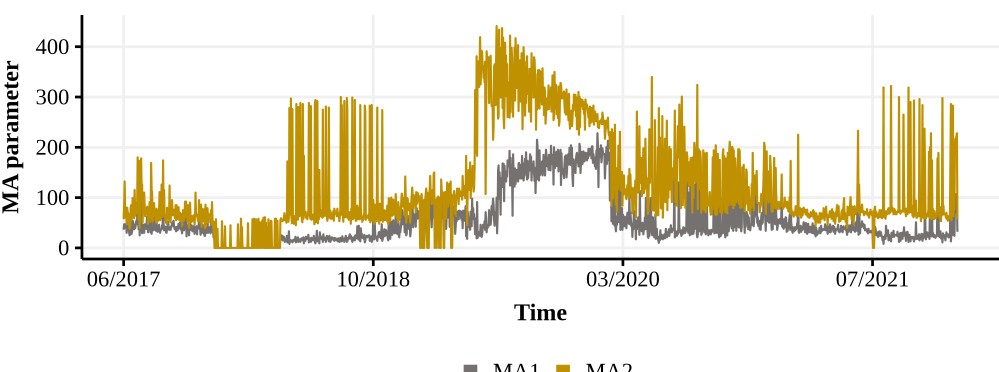

**Figure 5.** Evolution of the MA parameters.

For the case of the TR1 algorithm (Figure 4), we observe that from mid-2017, the MA1 parameters are around 50 days, and the MA2 parameters are around the moving average value of 100 days, remaining stable until the March 2020 period which coincides with the onset of the COVID-19 pandemic. Here, the MA2 moving average value rises around 200 days, with a higher level of variability.

Observing the TR2 trading algorithm series, we can see in mid-2017 that the MA1 series is centred around 40 days and the MA2 series at 65, changing at the beginning of 2018 with a significant increase in variability. In mid-2019, we observed a jump in the moving average series centred near 170 and 300 for MA1 and MA2, respectively. Similar to the series coming from the TR1 trading algorithm, we observed that the variability of the MA2 parameter selection for the start of the COVID-19 pandemic increases.

Using Equation (4) we can calculate the entropy for the time series distribution of the Bitcoin return and the joint distribution of the moving average parameters. Figure 6 shows the joint behaviour of both entropies. We can observe the existence of at least two distinct states that allow us to characterise the relationship between the entropy of the Bitcoin return series and the selection of the optimal parameters calculated from the moving averages, observing a state of low variability when observing the entropy of optimal trading parameters and low entropy of returns, and a state of the high variability of trading parameters and a high level of entropy of Bitcoin returns.

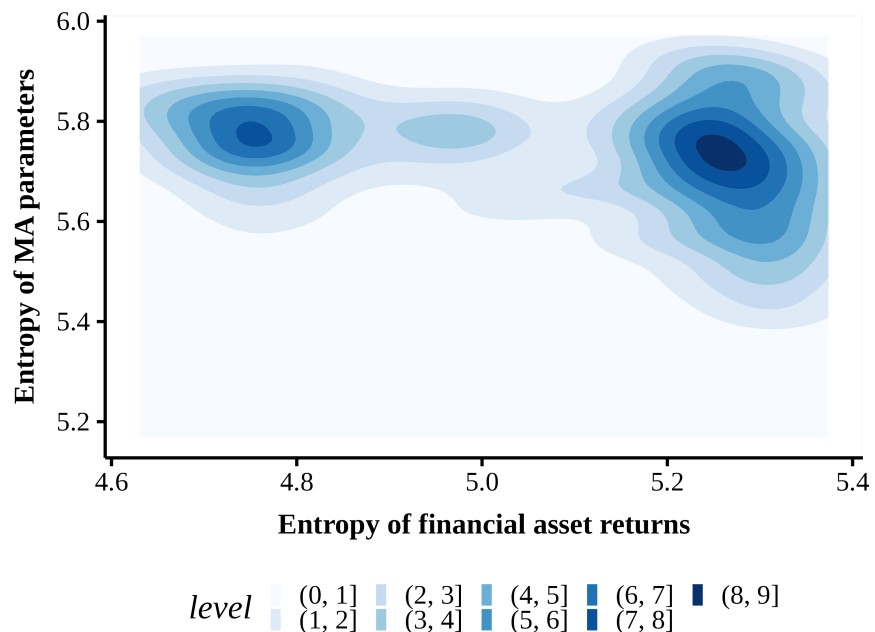

**Figure 6.** Joint entropy density of financial asset returns and MA parameters.

The reported value used to calculate entropy corresponds to the combination of $q_1$ and $q_2$ parameters that maximise the profit value, considering the best alternative between Algorithms 1 and 2. The general idea is to use the best parameter selection alternative, considering the best available algorithm alternative.

## 5. Discussion

Financial markets are currently experiencing a rapid and continuous influx of data, as well as the participation of a diverse group of investors with different investment horizons and feedback mechanisms. These factors give rise to complex phenomena such as market crashes or speculative bubbles. Studies have identified stylised facts, also known as complexity characteristics, that are present in most financial markets. The remarkable growth of the cryptocurrency market in recent years presents a remarkable opportunity to research its evolution over a short period of time.

The utilisation of high-frequency data makes it possible to generate advanced statistical analyses of variations in the values of cryptocurrency exchanges. After a few years after the introduction of the first cryptocurrency, Bitcoin, several thousand instruments have become available based on blockchain technology. The cryptocurrency market has progressed significantly from being a mere curiosity and niche interest among technology enthusiasts to emerging as a major global marketplace where substantial sums of money are actively traded on a daily basis.

This research primarily centres on the examination of the dynamics associated with cryptocurrency volatility and the volatility of parameter selection that defines optimal trading behaviour. The speculative nature of the cryptocurrency market contributes to the high volatility observed in the dynamics of cryptocurrency prices. A model that accurately captures the daily volatility of the high and low prices in a day can also effectively predict intra-day returns, valuable information for market speculators. To this aim, in the present research, we have used an approximate Bayesian computation technique and clustering methodology in the cryptocurrency markets, allowing us to optimise the trading strategy and the entropy of the Bitcoin price series. Thus, we observed at least two distinct states that characterise the relationship between the entropy of the Bitcoin return series and the selection of the optimal parameters calculated. Furthermore, we detect significantly similar behaviours to the results of our estimates in periods of high volatility, especially intensified by the COVID-19 outbreak. This pattern of behaviour suggests that the price formation process in cryptocurrency markets may be characterised by irrationality.

On the other hand, the observed instability in the parameters that define optimal trading behaviour is not necessarily an exclusive phenomenon of the cryptocurrency market. As part of future research, there is a need to explore this behaviour in other types of markets, such as the currency market, commodities market, etc.

The existence of clusters in the joint distribution between the entropy of Bitcoin returns and the joint distributions of optimal trading parameters implies that the choice of parameters and entropy of the overall trading system is time-varying, depending on the state of nature to which it belongs. This implies that proper modelling requires consideration of different states of nature within the processes governing the trading strategies.

## 6. Conclusions

The main contribution of this research provides substantial evidence that the Bitcoin market is connected to volatility in different states of nature through the optimal selection of parameters defining the trading strategy. This, coupled with the structural organization of the cryptocurrency markets, results in a complexity that is indistinguishable at the particular time series level. In contrast to traditional markets, the cryptocurrency markets exhibit less synchronization and slower information flows, which give rise to more frequent arbitrage opportunities. Consequently, many empirical studies have investigated the interlinkages, integrity, time-varying dynamics, and underlying fundamentals of the cryptocurrency markets.

Our findings have significant implications for academics, policymakers, and investors. It is crucial for investors to understand the impact of herding behaviour on the value of their portfolios, mainly when there are changes in the volatility structure of the markets. The COVID-19 pandemic may have played a role in exacerbating herding behaviour in financial markets, potentially attributed to policymakers' expansive monetary policies and resource misallocation. It is important that policymakers make well-informed decisions on the legal framework governing cryptocurrency markets and their integration with traditional financial markets.

Additionally, cryptocurrency exchange platforms should consider adjusting the minimum tick size to maintain market resilience and prevent trading activities from harming the market microstructure. These insights can also be useful for academics researching cryptocurrency market interlinkages, dynamics, and fundamentals.

The concept of entropy, initially introduced in thermodynamics, has been utilised in the field of finance for an extended period of time. This research uses the application of entropy in finance, and it focuses on identifying entropy states by selecting optimal parameters for the trading process using a Bayesian approach, showing the existence of at least two distinct states that affect the optimal selection of parameters.

Regarding limitations and future directions, we plan to address extreme conditions such as financial crises and orders of magnitude, utilising alternative entropy estimation methods, such as the histogram-based method, to evaluate the predictability of the financial series through the maximum entropy method, incorporating extreme volatility data influenced by social context. Additionally, further research may be conducted by scholars on the drivers of herding behaviour in cryptocurrency markets, with a focus on investor sentiment derived from social media, to gain a more comprehensive understanding of their trading dynamics.

**Author Contributions:** Data curation, H.d.l.F.-M., K.C.-J. and R.R.-T.; formal analysis H.d.l.F.-M., K.C.-J. and R.R.-T.; investigation, H.d.l.F.-M., K.C.-J. and R.R.-T.; methodology, H.d.l.F.-M., K.C.-J. and R.R.-T.; writing—original draft, H.d.l.F.-M., K.C.-J. and R.R.-T.; writing—review and editing, H.d.l.F.-M., K.C.-J. and R.R.-T. All authors have read and agreed to the published version of the manuscript.

**Funding:** Research work of H. de la Fuente-Mella was supported by grant Núcleo de Investigación en Data Analytics/VRIEA/PUCV/039.432/2020 from the Vice-Rectory for Research and Advanced Studies of the Pontificia Universidad Católica de Valparaíso, Chile.

**Data Availability Statement:** The data used to support the findings of this study are available from the corresponding author upon request.

**Conflicts of Interest:** The authors declare no conflict of interest.

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
