# Peer review of "A Stochastic Analysis of the Effect of Trading Parameters on the Stability of the Financial Markets Using a Bayesian Approach"

_mathematics, doi:10.3390/math11112527_

Round 1

Reviewer 1 Report

Dear authors,

I was happy to read your article. It is very interesting and there is a good scientific soundness in your work.

There is suggestion about entropy testing that I would like to point out.

Shannon entropy is far surpassed in many similar studies, especially if we consider cryptocurrencies and Bitcoin. Shannon function is very suitable to measure uncertainty related with a random variable contained in another random variable. Its a good nonparametric method for measuring casual information transfer between systems in directional flow.

However, Renyi transfer entropy seems to be more appropriate for such study. It will use shuffled version of time series and discover weighted parameter q.

Best regards

Reviewer 2 Report

This paper proposes a stochastic analysis of the effect on the stability of trading parameters in financial markets using a bayesian approach. The topic is interesting but there are some issues to be considered.

1. This paper has a significant similarity score, which raises concerns about its originality and contribution to the field.

2. There are several typo and grammatical errors in the paper.

For example:

- In the title of this paper, article 'An' should be replaced by 'A'.

- In line 22-24, “The Chinese government’s mining ban and restrictions imposed in other countries were due to environmental impact. Cryptocurrency mining has a huge citemeynkhard2019energy, goodkind2020cryptodamages, nanez2021cryptocurrency im- pact in Asia, Latin America, and Africa.”, the citations format should be revised.

- The word "Bitcoin" should be capitalized when referring to the specific cryptocurrency. However, in several paragraphs, “bitcoin” is used instead of “Bitcoin”.

3. This paper is lacking in result comparison with existing algorithms.

Reviewer 3 Report

The article "An stochastic analysis of the effect on the stability of trading parameters in financial markets using a Bayesian approach" developed a mathematical model for the stability of cryptocurrency financial markets.

The structure of the article is correct, the literature review is sufficient, but the analysis should be better described.

The summary does not reflect the essence of the article, the final conclusions are too general. The Authors' own contribution was not indicated, nor were the limitations and innovations of the developed model given.

The drawings are hard to read and not sufficiently explained.

The algorithm itself should be enriched with comments so that it can be transferred to other studies.

Reviewer 4 Report

Here are some suggestions for improving the manuscript:

1.          The abstract should provide a concise summary of the manuscript and its results. Please revise the abstract to reflect this.

2.          Clarify your research question and objectives: While you have provided a detailed background and methods section, it is important to clarify your research question and objectives more explicitly in the introduction. This will help readers to better understand the scope of your research and what you aim to achieve.

3.          Provide more details on the data: While you have described the data used in your study, it would be helpful to provide more details on the sources of the data, including how it was collected and any limitations or biases that may be present.

4.          Please check the manuscript for lowercase errors, such as line 132 on page 3 where "generalized ARCH (GARCH) model" is written in lowercase.

5.          Improve the organization of your manuscript: Your manuscript would benefit from better organization. Consider reorganizing your sections to make the flow of information more logical and easier to follow.

6.          Provide more context for your findings: While you have presented your findings in a clear and concise manner, it would be helpful to provide more context for readers unfamiliar with the field. Consider including more background information on the cryptocurrency market and how your findings contribute to our understanding of this market.

7.          Discuss the limitations of your study: While you briefly touch on the limitations of your study in the conclusion, it would be beneficial to discuss them in more detail throughout the manuscript. This will help readers to better understand the scope of your research and the potential implications of your findings.

 Overall, the manuscript requires significant work before it can be considered for publication. Please address these suggestions to improve the quality of the manuscript.

Round 2

Reviewer 2 Report

I appreciate the efforts made by the authors to revise the paper based on the previous comments, which has led to an improvement in its quality. However, there are some remaining issues that need to be addressed before the paper can be accepted.

1. The paper has been revised, but its similarity score remains at 24%, with 7% from MDPI sources. Please review the content and reduce the similarity score to below 20%."

2. The paper has been revised to address the errors given as examples in the previous comments. However, I kindly request the authors to review the entire paper, not just the previously mentioned errors. The current version still contains several typos and grammatical errors. Therefore, I recommend a thorough review of the paper to identify and correct any remaining errors. Please provide a comprehensive list of all errors found and resolved in the revised version."

3. In the revised version, in section 4.1's first paragraph, the authors have added the following sentence: “However, the data used for the training 260 corresponds from May 2013 to May 2017, leaving data available for tracking and testing 261 the algorithms from June 2017 to December 2021.” However, the paper does not provide any information about the performance of the training and testing. Therefore, I recommend including the performance results and comparing them to existing methods in the literature to highlight the contribution of this work.

Reviewer 4 Report

The authors have completely improved the manuscript following all my comments. The paper can be now accepted for publication.

Author Response

Thank you very much for your comments.

Round 3

Reviewer 2 Report

I appreciate the efforts made by the authors to revise the paper based on the previous comments.